# Development of a Performance-Enhanced Hybrid Magnetorheological Elastomer-Fluid for Semi-Active Vibration Isolation: Static and Dynamic Experimental Characterization

**DOI:** 10.3390/ma15093238

**Published:** 2022-04-30

**Authors:** Abdelrahman Ali, Ayman M. H. Salem, Asan G. A. Muthalif, Rahizar Bin Ramli, Sabariah Julai

**Affiliations:** 1Department of Mechanical and Industrial Engineering, College of Engineering, Qatar University, Doha P.O. Box 2713, Qatar; abdelrahman.ali@qu.edu.qa; 2Department of Mechanical Engineering, Faculty of Engineering, University of Malaya, Kuala Lumpur 50603, Malaysia; 17220126@siswa.um.edu.my (A.M.H.S.); rahizar@um.edu.my (R.B.R.); sabsz@um.edu.my (S.J.)

**Keywords:** hybrid materials, magnetorheological elastomers, magnetorheological fluids, vibration isolation, transmissibility factor

## Abstract

Magnetorheological elastomers (MREs) are a class of emerging smart materials in which their mechanical and rheological properties can be immediately and reversibly altered upon the application of a magnetic field. The change in the MRE properties under the magnetic field is widely known as the magnetorheological (MR) effect. Despite their inherent viscoelastic property-change characteristics, there are disadvantages incorporated with MREs, such as slow response time and the suspension of the magnetic particles in the elastomer matrix, which depress their MR effect. This study investigates the feasibility of a hybrid magnetorheological elastomer-fluid (MRE-F) for longitudinal vibration isolation. The hybrid MRE-F is fabricated by encapsulating MR fluid inside the elastomer matrix. The inclusion of the MR fluid can enhance the MR effect of the elastomer by providing a better response to the magnetic field and, hence, can improve the vibration isolation capabilities. For this purpose, an MRE-based coupling is developed, and isolation performance is investigated in terms of the linear transmissibility factor. The performance of the hybrid MRE-F was compared against two different MRE samples. The results show that further enhancement of MR-effect in MREs is possible by including MR fluid inside the elastomer. The hybrid MRE-F exhibited better stiffness change with the current increase and recorded the highest value of 55.911 N/mm. The transmissivity curves revealed that the MRE-F contributed to a broader shift in the natural frequency with a 7.2 Hz overall shift at 8.9 mT. The damping characteristics are higher in MRE-F, recording the highest percentage increase in damping with 33.04%. Overall, the results reveal the promising potential of hybrid MRE-F in developing MRE-based coupling for longitudinal vibration isolation.

## 1. Introduction

The desire for a better and enhanced performance of engineering devices has led to a growing demand for advanced materials. Smart materials are among those that have received the utmost attention from many researchers. Smart or advanced materials can be controlled by external stimuli such as heat and electrical or magnetic fields [1].

Magnetorheological (MR) materials are smart composites with wide capabilities in various industrial applications. The rheological and mechanical properties of MR materials can be alerted under the influence of an external magnetic field. Stiffness, damping, yield and shear stress, and dynamic moduli are among the properties that can be tuned or controlled when an external magnetic field is applied to MR materials [2]. MR materials typically consist of a magnetic particle that is dispersed and suspended in a carrier medium. Other analogues to MR materials, such as electrorheological (ER) materials, have also been proposed in many studies. However, MR materials are superior to ER materials due to their higher performance characteristics. Depending on the medium employed, MR materials can include MR foam, MR gels, MR fluid (MRF), and MR elastomer (MRE) [3,4,5,6]. The two main branches of the MR materials that have been mostly adopted in research are MRFs and MREs. In MRFs, magnetic particles are dispersed in a fluid medium such as silicone oil, mineral oil, or petroleum-based oil. These materials have been successfully adopted in developing MR brakes, mounts, and suspensions [7]. The wide application of MRFs is due to their easy control, fast field response, and noiseless operation [8]. Nevertheless, studies have reported some disadvantages incorporated with MRFs that limit their application. Such drawbacks include oil medium leakage, contamination, and magnetic particles sedimentation [9]. The sedimentation of the magnetic particles in the oil is caused by the density difference between the magnetic particles and the oil. As for MREs, they consist of magnetic particles dispersed in an elastomeric matrix material. Figure 1 shows a typical response of the magnetic particles in MREs when subjected to a magnetic field. Under the application of an external magnetic field, the interconnection between the magnetic particles increases, which can change the viscoelastic behavior of MREs [10]. This can be considered as the solid equivalent behavior to MRFs. Therefore, the solid-state of MREs can overcome sedimentation issues in MRFs. Although, various solid-equivalent materials to MRFs are being continuously developed to eliminate the sedimentation issue in MRFs, MRE remains the most widely adopted because of its superior characteristics. MREs offer change in electrical resistance and capacitance in addition to its rheological and mechanical properties.

MREs have gained considerable attention because of their distinct advantages over MRFs. MREs are essentially used in vibration control technologies owing to their inherent field dependence and controllable mechanical properties such as stiffness and damping. Thereby, MRE-based dynamic vibration absorbers and vibration isolators have been developed to attenuate the detrimental vibration levels that can cause damage and deterioration of the machine components [11,12]. The conventional vibration absorbers and isolators can only operate within a narrow frequency band. Hence, it is difficult to tune vibrations occurring at high frequencies. MREs emerged as a viable solution that can provide the ability to control the frequency of vibration control devices. For this reason, MRE-based vibration absorbers and isolators are successfully employed to eliminate external vibration excitations at high-frequency bands. Ginder et al. [13] developed an active controllable MRE absorber, while Deng et al. [14] proposed an MRE-based absorber that can tune frequencies in the shear mode. Leng et al. [15] developed a mixed-mode MRE-based isolator that can operate in the shear and squeeze combination. Syam et al. [16] proposed a semi-active vibration isolator using MRE for drilling systems to isolate torsional vibrations. Salem et al. [17] developed metamaterial MRE-based coupling with different activation modes to isolate vibrations at broadband frequencies.

As for the fabrication of MREs, various techniques have been extensively adopted and reported in the literature. Conventionally, MREs are fabricated by mixing micron-sized magnetic particles into a typical silicone elastomer polymeric matrix and curing in a casting mold. The magnetic particles are randomly distributed or aligned to a specific direction in the polymeric matrix, and they are categorized as isotropic and anisotropic MREs, respectively [18,19]. During the cross-linking process, a magnetic field is required to align the magnetic particles in a specific direction. Thus, in anisotropic MREs, the magnetic particles form chains along the direction of the magnetic field and are suspended upon curing [20]. On the other hand, hybrid MREs have been developed by adding MRF to the elastomeric matrix, which can be encapsulated and oriented in a specific direction within the elastomer. Bisola et al. [21] proposed the 3D-printing of hybrid MRE material where multi-material printing was employed. In this process, MRF filament is encapsulated inside the rubber matrix by a non-vulcanizing suspension process. Qi et al. [22] fabricated a 3D-printing plastic filament mixed with CIPs using fused filament fabrication. The CIP and plastic mixes were printed and encapsulated with silicone rubber by manually pouring the rubber. The addition of the MRF can eliminate the issues incorporated in MREs, such as the slow response time due to the suspension of the magnetic particles. Hybrid MREs have been employed to develop vibration isolators to expand their modes of operation. Xeng et al. [23] proposed a novel semi-active absorber based on MRE and MRF, which are working in shear and squeeze modes, respectively. Sun et al. [24] designed a hybrid MRE isolator to improve stiffness and damping tunability with an MRE isolator and MRF damper.

In this study, an investigation is conducted on the performance of hybrid magnetorheological elastomer-fluid (MRE-F) in longitudinal vibration isolation. The investigation is intended to examine the stiffness and damping characteristics of the hybrid MRE-F material and evaluate its performance against other MRE materials. MRE-F is fabricated by encapsulating a mix of silicone oil and magnetic particles directly within the MRE structure. The second material used for comparison included silicone oil solely encapsulated (MRE-S), while the third material represented is a conventional MRE. The hybrid materials were fabricated using a customized casting mold fabricated using 3D printing, which allows fluid encapsulation within the structure of the fabricated MRE. Hybrid MRE-based coupling is developed to investigate the performance of the three models in attenuating longitudinal vibrations. The vibration level is quantified by measuring the transmissibility of the system. The shift in the system’s natural frequency and changes in damping are measured and reported.

The paper is organized as follows. Section 2 presents the mathematical modeling of the system, and Section 3 presents the experimental setup developed to conduct the investigation. An analysis of the results and the system response is presented and discussed in Section 4. The summary and concluding remarks are highlighted in Section 5.

## 2. Mathematical Modelling

MRE-based coupling consists of upper and lower coupling hubs, coils, and an MRE layer, as shown in Figure 2. Coupling is embedded with a vibration shaker attached to the input shaft holder along with an impedance sensor to measure the input force and acceleration.

Output acceleration is measured using an accelerometer attached to the upper coupling hub. The MRE layer is composed of the elastomer and encapsulated fluid. The system shown in Figure 2 is used to derive the mathematical model based on a fundamental equation of motion. The exact mathematical modelling of a physical system leads to complicated nonlinear differential equations. Therefore, the modelling of the system is assumed to be linear to eliminate nonlinear terms in the governing equation or approximate them by linear terms [25]. The system is a single degree of freedom, with an input displacement, y, and output angular displacement, x. The equation of motion is developed to obtain the system performance and is described as follows:(1)$$m\ddot{x}+c_{mre}\left(\dot{x}-\dot{y}\right)+k_{mre}\left(x-y\right)=0$$
where m is the absorber mass, and $k_{mre}$ and $c_{mre}$ are the stiffness and damping coefficient of the hybrid MRE, respectively. By applying the Laplace transform, the transfer function of the system is obtained as follows.
(2)$$\left[ms^{2}+c_{mre}s+k_{mre}\right]X\left(s\right)=\left[c_{mre}s+k_{mre}\right]Y\left(s\right)$$

Hence, the transmissibility factor equation is obtained by transforming Equation (2) from the *s*-domain to the frequency domain (ω-domain) by replacing s=jω.
(3)$$\frac{Output\left(j\omega \right)}{Input\left(j\omega \right)}=\frac{Y\left(j\omega \right)}{X\left(j\omega \right)}=\frac{c_{mre}j\omega +k_{mre}}{m{\left(j\omega \right)}^{2}+c_{mre}j\omega +k_{mre}}$$

Taking the magnitude of Equation (3) results in transfer function (TF), where ω is the natural frequency in (rad/s).
(4)$$\frac{Output\left(\omega \right)}{Input\left(\omega \right)}=TF=\left|\frac{Y\left(\omega \right)}{X\left(\omega \right)}\right|=\sqrt{\frac{{\left(c_{mre}\omega \right)}^{2}+{\left(k_{mre}\right)}^{2}}{{\left(c_{mre}\omega \right)}^{2}+{\left(k_{mre}-m\omega^{2}\right)}^{2}}}$$

The assumption of linear modeling has some limitations manifested in the narrow frequency band and lower robustness than nonlinear systems [26]. Nonlinear isolators can effectively reduce the vibration level with lighter mass and a wider frequency band of vibration attenuation. Nonetheless, linear modeling is adopted in this study to avoid the complexity and errors associated with nonlinear modeling [27].

## 3. Preparation of the Hybrid MRE Samples

### 3.1. Casting Mold Design

The MRE samples were fabricated and cured using a customized casting mold. The mold was manufactured by 3D printing using polylactic acid (PLA) 3D-printing filament. MRE samples with internal cavities were cast using the mold shown in Figure 3. The mold consists of three plates, as shown in Figure 3a. The lower plate has the first two tongues that will form the grooves in the elastomer. A centered extrusion in the lower plate holds a Styrofoam piece from below. Similarly, the upper plate has the other two tongues for the upper grooves and extrusion to hold the Styrofoam piece from above.

The middle plate defines the overall length of the casted samples. The mold was designed with a degree of flexibility in design modifications. A longer MRE sample can be easily fabricated by only changing the height of the middle plate. Figure 3b,c show the casting mold assembly.

### 3.2. Fabrication of the Hybrid MREs

The fabrication process of the hybrid MRE samples is analogous to the conventional fabrication process. The fabricated MRE consists of two main components: silicone elastomer and carbonyl iron particles (CIPs). The silicone used in the fabrication process is Elite Double 32 Fast from Zhermack, which has high elastic recovery and resistance to stretching and tearing. This type of silicone does not require mixing in a vacuum because of its high fluidity. It consists of a silicone base and catalyst mixed at a 1:1 ratio and its curing time is 20 min. The sedimentation of the magnetic particles after mixing is minimized due to the fast-curing time of the silicone. As for CIPs, it is formulated by the thermal decomposition of iron pentacarbonyl ${\;\mathrm{FeCO}}_{5}$. CIPs are easy to magnetize and have high saturation and demagnetization characteristics, making them suitable for MRE fabrication. The CIP type used in this study is SQ-I developed by BASF. This type of CIPs has high magnetic saturation and high magnetic permeability. It is also coated with silicone dioxide ${\mathrm{SiO}}_{2}$ to increase compatibility with the elastomer matrix and reduce sedimentation [28]. The stiffness and hardness of the MRE samples are highly influenced by the addition of the magnetic filler. Soft MRE samples have better elastic behavior and are able to isolate more vibrations. However, if the rubber is too soft, it might not be able to recover from deformity, and it will reduce the effectiveness of the vibration isolation. Therefore, CIPs are added with a 10% volume fraction which is an intermediate between sufficient magnetization and passive stiffness. The magnetic particles used in this study are polydisperse with varying particle size distribution. This is essential to have a better MR effect due to particle packing in chain-like aggregates and increased magnetic dipole interaction with adjacent particles [29,30].

The fabrication process of the MRE samples includes three steps: Mixing, curing, and the addition of the fluids. The MRE fabrication process is clearly presented in Figure 4. Firstly, CIPs were added to the silicone base and mixed. The mass required for the CIPs is calculated using the following formula:(5)$$m_{CIPs}=\left(\rho_{CIPs}\times \frac{\pi r^{2}h}{1000}\right)\;\times 10\%$$
where r is the radius, h is the height, and $\rho_{CIPs}$ is the density of the magnetic particles. After both parts are mixed adequately, a silicone catalyst was added and stirred. Then, the mix was poured into the casting mold that contained the Styrofoam piece. The mold is then covered with the upper plate. Excess MRE liquid can flow out of the mold through two risers located at the upper plate. The risers prevent cavities from forming due to shrinkage during solidification.

Additionally, the MRE formed at the risers is used as plugs to close the grooves created by the extrusions that hold the Styrofoam. After the sample is cured, chloroform was used to flush the MRE sample inside to dissolve the Styrofoam. MR fluid was injected into the cavity created by the Styrofoam for the first MRE sample. This fluid was created by mixing CIPs with 30% volume fraction silicone oil. The second sample contained pure silicone oil in the internal cavity, while the third was hollow. Table 1 compiles the material properties of the silicone elastomer and the CIPs, while Table 2 summarizes the samples that are considered in the investigation.

### 3.3. Development of the Hybrid MRE-Based Coupling

Hybrid MRE-based coupling consists of four main components: upper and lower coupling hubs, coils, and the hybrid MRE layer, as shown in Figure 5a. The overall dimensions of the coupling are shown in Figure 5b. The coupling is designed to wind coils around the grooved section of the coupling hubs. Therefore, coupling is used as an electromagnet that supplies the electrical current to the coil and, in turn, induces the magnetic field to the MRE layer. Electromagnetic coupling is controlled to have opposite magnetic polarity; as a result, the magnetic flux is directed through the MRE layer. The magnetic flux density is recognized as follows [31]:(6)$$B=\mu_{o}\;\mu \;N\;I$$
where $\mu_{o}=4\pi \times 10^{-7}\frac{H}{m}\;$ is the magnetic permeability of the vacuum, μ is the relative permeability of the core, N is the number of turns in the coil, and I is the excitation current.

## 4. Experimental Setup and Methodology

The performance of the hybrid MRE-based coupling is investigated by measuring the transmissibility of three MRE samples. An experimental setup was created to perform the longitudinal vibration test. The setup is shown in Figure 6, while the schematic is represented in Figure 7. The experimental setup consists of vibration steel mounted on vibration absorbing pads. The table ensures the stability of the structure while performing the test and minimize the vibration noise transmitted to the sensors. A permanent magnet vibration shaker (DS-PM-100) from Dewesoft^®^ is used as the exciter. The shaker is equipped with an integrated power amplifier and controlled by data acquisition. The impedance sensor was attached at the shaker’s excitation port to measure the input force and acceleration. MRE-based coupling was installed on top of the impedance sensor, while an accelerometer was mounted on top of the coupling. Dewesoft Sirius data acquisition system (DAQ) w used to perform a sine sweep test for a frequency range from 0 to 250 Hz. DAQ is equipped with a dynamic signal analyzer to analyze the signal and plot the frequency response function (FRF), indicating transmissibility curves. A DC/AC regulated power supply is used to supply electrical currents to electromagnetic coils. The coils are supplied with electric current ranging from 1 to 3 A. A Tesla meter was used to measure the magnetic field intensity of the electromagnetic coils. From the transmissibility curves, the half bandwidth theory can be applied to obtain stiffness and the damping coefficient of the MRE layer. The damping ratio *ζ* can be expressed as described by the Q factor as follows [32].
(7)$$\zeta =\frac{1}{2Q}$$

The Q factor can be expressed by the following:(8)$$Q=\frac{f_{n}}{f_{2}-f_{1}}$$
where $f_{n}$ is the resonance frequency, and $f_{1}$ and $f_{2}$ are the half-power frequencies measured at 0.707 Q or at $\frac{1}{\sqrt{2\;}}TF$. The stiffness of the MRE can be obtained from the frequency domain function. The stiffness $k_{mre}$ of the MRE is obtained from the expression:(9)$$\omega_{n}=\sqrt{\frac{k_{mre}}{m}}$$
where $\omega_{n}$ is the natural frequency of the system in rad/s, and m is the mass of the upper coupling hub. The damping coefficient c of MRE can be obtained in terms of the stiffness $k_{mre}$, and the damping ratio ζ  by using critical damping $c_{c}$ as follows.
(10)$$c_{c}=2\sqrt{k_{mre}m}$$
(11)$$c=\zeta c_{c}$$

A static compression test was performed to determine the behavior of the hybrid MRE samples under compressive loads. The test was conducted using INSTRON 5585H universal testing machine (UTM). A schematic diagram of the UTM shown in Figure 8 depicts compression test procedure. MRE coupling is placed between two wooden compression plates to ensure that the magnetic flux is directed to the MRE layer and not scattered due to metal surface interference. The test is performed on all three MRE samples where the coupling is exposed to a continuously increasing compressive load. The maximum compression is applied is 12 mm with a compression rate of 3 mm/min. This ensures the adequate compression of the MRE samples, which contain fluid such as the MRE-F and MRE-S. The intensity of the magnetic field is varied using the regulated power supply and the load-displacement curves are measured to acquire the MRE layer stiffness. The induced MR-effect of the hybrid MRE samples is evaluated from the change in the MRE modulus retrieved from the curves.

## 5. Results and Analysis

The magnetic field intensity produced inside the electromagnetic coils was measured at different current levels using the Tesla meter. The opposite magnetic polarity between the upper and lower electromagnets ensured that the magnetic field is focused on the MRE layer. The maximum magnetic flux is accumulated around the surface of the tongue-groove connection between the MRE layer and the coupling hubs. While sufficient magnetic strength is measured along the length of the MRE layer. The maximum magnetic filed generated around the MRE is recorded and plotted in Figure 9. It is observed that the magnetic flux supplied by the electromagnetic coils increases with the current’s supply. The maximum recorded magnetic filed value is 8.9 mT when the current supply is at maximum of 3 A. The magnetic flux generated around the MRE is restricted by the maximum current provided by the regulated power supply to the coils. The magnetic field intensity measurements are essential to ensure that the MRE layer is subjected to sufficient magnetic flux to achieve the optimum MR effect.

### 5.1. Static Compression Test

The loads against the displacement curves recorded by UTM are plotted and shown in Figure 10. The test was performed on MRE samples to investigate the change in their stiffness with the applied magnetic field. The stiffness of each MRE sample was obtained by measuring the slope of the load-displacement curve. Curve fitting using the line of best fit with the coefficient of determination above 90% indicates a good linear correlation between the force and displacement. The stiffness values of MRE-F, MRE-S, and MRE-H at different current levels are summarized in Table 3. The maximum stiffness induced by the magnetic field was found in the case of MRE-F with a value of 55.911 N/mm. As for the MRE-S, it contributed to the second highest stiffness with a maximum value of 53.592 N/mm, while MRE-H recorded the lowest stiffness value with 30.948 N/mm.

Analyzing the results obtained from the static compression testing, the following conclusions can be made:It was observed that the stiffness of the MRE samples increased when a higher current is supplied to the electromagnetic coils. Increasing the current supply resulted in an increase in the magnetic field applied to the MRE samples by the electromagnetic coils. As a result, the inter-particle connections between the magnetic particles suspended within the elastomer is enhanced, which increased the stiffness of the MRE layer.In the presence of the magnetic field, the compression of MRE could cause magnetic particles to move from a minimal energy state, which requires additional work. This work increases with the applied magnetic field, resulting in field-dependent stiffness or modulus [33].The silicone oil mixed with CIPs in the case of MRE-F starts behaving similarly to a semi-solid as a function of field intensity in the presence of magnetic field. For this reason, the stiffness of MRE-F is higher under the influence of magnetic field because of the additional chains formed by the magnetic particles suspended within the fluid.When the magnetic field is removed, the mixed silicone oil behaves like a normal carrier fluid again. This indicates the reversible rheological behavior of the fluid suspended inside MRE-F.The MRE-F has shown a better stiffness change in response to the magnetic current, which indicates an enhanced MR-effect. Figure 11 shows the total percentage change in stiffness of the MRE samples at different current increments. The percentage increases in stiffness for MRE-F, MRE-S, and MRE-H when they are shifted from passive to the active state at 1 A current increment were 0.73%, 0.61%, and 0.56%, respectively. At a higher current increment from (2 to 3 A), the percentage increase is the highest in the case of the MRE-F, with a value of 1.51%. The maximum increase in stiffness is achieved by MRE-F at 3 A current increments, with a total of 2.45% increases. Overall, the results of the compression tests indicated that MRE-F has a better response to the magnetic field.

### 5.2. Relative MR-Effect

The MR-effect is a crucial parameter for evaluating the performance of hybrid MRE samples. The relative MR-effect is measured from the ratio of the magneto-induced modulus and the initial modulus and can be evaluated by the following equation:(12)$$\mathrm{Relative}\;\mathrm{MR}-\mathrm{effect}=\frac{E_{max}-E_{0}}{E_{0}}\times 100\%$$
where $E_{0}$ is the zero-field modulus, and $E_{max}$ is the maximum modulus when the magnetic field is applied to MRE. The term $E_{max}-E_{0}$ is referred to as the magneto-induced modulus. The modulus is evaluated directly from the force-displacement curves using the following equation:(13)$$E=\frac{FL_{0}}{Ad}$$
where F is force, $L_{0}$ is the initial length, A is the cross-sectional area, and d is the displacement. The MR-effect is influenced by several factors, including magnetic field strength, CIPs content, particle size and distribution, and the type of the matrix material [34]. In this study, the MR-effect of the three hybrid MRE samples can either be influenced by the magnetic field or the particles size distribution since the CIPs’ content and the MRE matrix are maintained for all samples. The magnetic particles used in this study are polydisperse possessing a wide particle size distribution, which can impact the performance of MRE samples. Several studies have reported the performance of MREs based on the filler size and distribution within the MR matrix. The MR-effect is expected to be higher for MRE with a mixed-size of iron particles. This is because, for any cubic filled up with a large particle, there was an extra small size particle next to it [35]. This increased the magnetic dipole interaction with the adjacent particle. MR materials with large particle sizes have a higher MR-effect than small particle sizes. The relative MR-effect of the three MRE samples is evaluated under different magnetic fields. The zero-field modulus, magneto-induced modulus, and MR effect data are summarized in Table 4.

The following conclusions can be made based on the relative MR-effect results:
It was observed that the MR-effect increased with the applied magnetic field, due to the slight increase in magneto-induced modulus. The enhancement of the MR-effect with the magnetic field has been reported in several studies [36,37,38]. The relative MR-effect under different magnetic field intensity levels of the hybrid MRE samples is shown in Figure 12.A higher MR-effect is observed in the case of MRE-F. This is because of the fast response of the magnetic particles dispersed in the silicone oil. These magnetic particles can freely move within the carrier fluid and form chain-like structures parallel to the magnetic field lines in addition to those formed by particles suspended within the elastomer.It is observed that the MR-effect in MRE-H is very close to that of MRE-S, with the former having a higher MR-effect at certain magnetic field levels. This is because of having a lower zero-field modulus compared to MRE-S. Therefore, a higher MR effect can be depicted in MRE, which has softer structure.

**Figure 12 materials-15-03238-f012:**
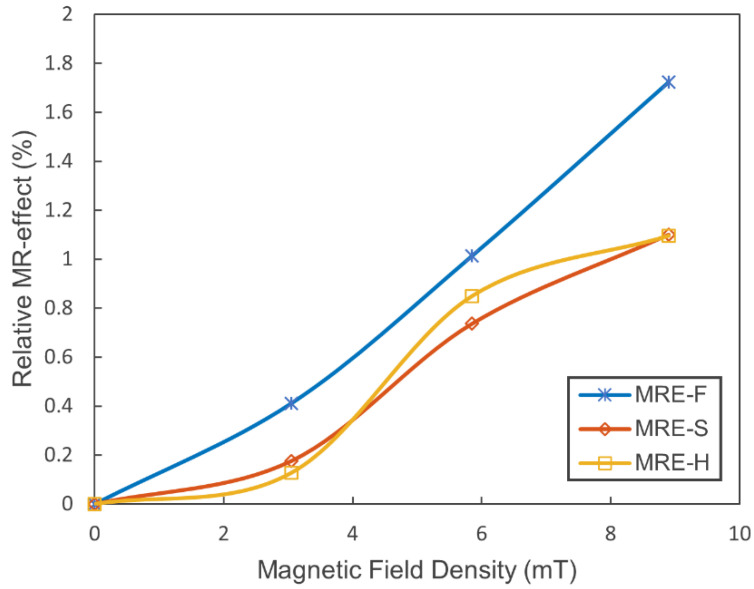
The relative MR-effect of the hybrid MRE samples under different magnetic fields.

### 5.3. Dynamic Vibration Test

Figure 12 shows the linear transmissibility curves of the three MRE samples at different current values where $g_{in}$ and $g_{out}$ are the input and output acceleration, respectively. All three samples have shown a linear transmissibility characteristic similar to previous studies reported in the literature [17]. The resonant frequency occurred in the range of 50−100 Hz for all samples where transmissibility reached maximum values. Therefore, the low-frequency range is the most critical range for this vibration test, similarly to most mechanical equipment that suffers high vibration amplitudes at low frequencies [39]. At higher frequency ranges above 100 Hz, the value of the transmissibility factor recorded the minimum values, which indicate the isolation region of the linear isolator. From Figure 13, it can be observed that the transmissibility peaks shift to the right as the current increases. This indicates that the natural frequency occurs at a higher frequency due to the increase in MRE stiffness. For MRE-F, the resonant frequency at the passive state, i.e., when no magnetic field was applied, occurred at 64.8 Hz. When the current was further increased to 1 A (3.05 mT), the resonant peak shifted to 65.6 Hz, indicating a 0.8 Hz shift in natural frequency. At higher current increments, the natural frequency shifted further to 72 Hz at 3 A (8.9 mT). The highest natural frequency shift was achieved by the MRE-F with a 7.2 Hz overall shift. This is because of the superior MR effect of the MRE-F sample since it has higher stiffness characteristics at higher current values. As for MRE-S, the maximum frequency shift was 6 Hz (from 66.4 Hz at 0 mT to 72.4 Hz at 8.9 mT). While the MRE-H sample contributed to the lowest frequency shift with only 2 Hz. Figure 14 depicts the percentage increase in the natural frequency of three samples at different current values.

The maximum possible reduction percentage for three MRE samples is compared in Figure 15, where zero reduction means that the response is amplified. It can be observed that the isolation regions for all samples begin at frequencies above 100 Hz. Hence, hybrid MRE-based coupling can attenuate vibrations at a frequency range of 100–250 Hz. MRE-F, MRE-S, and MRE-H reduction percentages are 82.14%, 82.5%, and 80.56%, respectively. The narrow frequency band of isolation is due to using a coupling with a single MRE layer. The isolation band can further be increased by using multiple MRE layers in the coupling. Furthermore, a reduction in the transmissibility peaks was observed as the MRE samples were subjected to higher current values. For MRE-F, the amplitude of the resonant peak dropped from 9.1 to 7.7 as current increased from 0 to 3 A. Similarly, the MRE-S sample recorded a decrease in the transmissible factor value at the resonant frequency as it dropped from 9.5 to 8.9 at the same current increment. The decreased transmissibility factor at resonance was not depicted for all MRE samples. The hollow MRE-H sample recorded an amplitude increase at resonance with increasing current. The observed results show that the damping characteristics of the MRE samples are different. The variation in the damping coefficients of MRE-F, MRE-S, and MRE-H were investigated and calculated by the half-power bandwidth method using Equations (7) and (11). The calculated damping coefficients for the three samples at different current levels are summarized and reported in Table 5. The upper coupling hub is the mass of the absorber used to calculate the critical damping coefficients. The percentage increase in the damping coefficients with the current levels is shown in Figure 16. It can be observed that the damping coefficients of MRE-F and MRE-H witnessed a slight increase with the increasing magnetic field without specific variation.

MRE-F recorded the highest percentage increase in the damping with a value of 33.04%. This states that the damping characteristics of MRE-F are superior to that of MRE-S and MRE-H. The findings agree with the literature as the damping characteristics of MRE are minimally affected by the magnetic fields.

## 6. Conclusions

This work presents the design, fabrication, and working principle of a hybrid MRE-based coupling for longitudinal vibration isolation. Three MRE samples were fabricated, differing in the type of fluid encapsulated within the cavity of the MRE. MRE-based coupling was experimentally investigated for the potential development of semi-active longitudinal vibration isolation. Vibration and compression tests were performed on coupling to investigate the stiffness and damping characteristics of each of the three samples. The results revealed that increasing the magnetic field intensity across the MRE samples can increase their stiffness. It was found that the MRE-F sample has the highest MR effect as it contributed to the greatest change in stiffness. The vibration test results showed that the natural frequency could be shifted to higher values by increasing the electrical current supplied to electromagnetic coils. MRE-F contributed to the highest frequency shift with a 7.2 Hz overall shift at 8.9 mT. The damping coefficients of the MRE samples showed a slight increase with the applied current without specific variations, with MRE-F recording the highest increase in damping with a value of 33.04%. This states that the damping characteristics of MRE-F are superior to that of MRE-S and MRE-H. Overall, the results reveal the promising potential of hybrid MRE-F in developing MRE-based coupling for vibration isolation.

## Figures and Tables

**Figure 1 materials-15-03238-f001:**
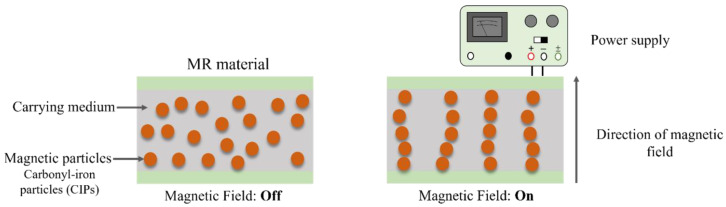
The distribution of the magnetic particles in MREs with and without magnetic field.

**Figure 2 materials-15-03238-f002:**
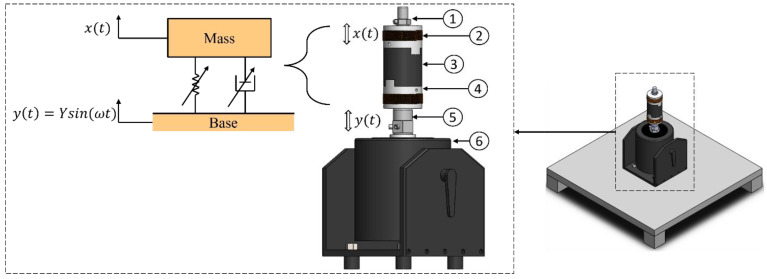
Model representing the base excitation of the hybrid MRE coupling: (1) accelerometer; (2) coils; (3) hybrid MRE; (4) coupling hub; (5) impedance sensor; (6) vibration shaker.

**Figure 3 materials-15-03238-f003:**
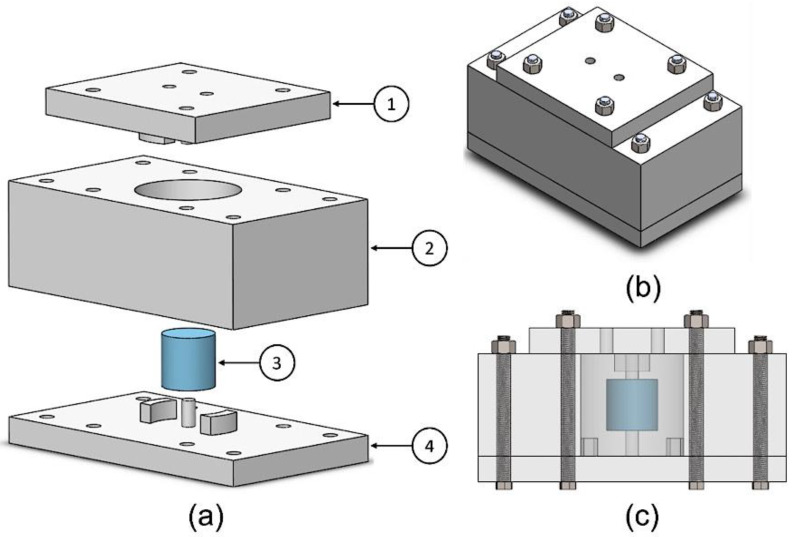
Hybrid MRE material mold design; (**a**) exploded view: 1—upper plate; 2—middle plate; 3—Styrofoam piece; 4—lower plate. (**b**) Isometric view of the assembled mold. (**c**) Front view showing the position of the Styrofoam inside the mold.

**Figure 4 materials-15-03238-f004:**
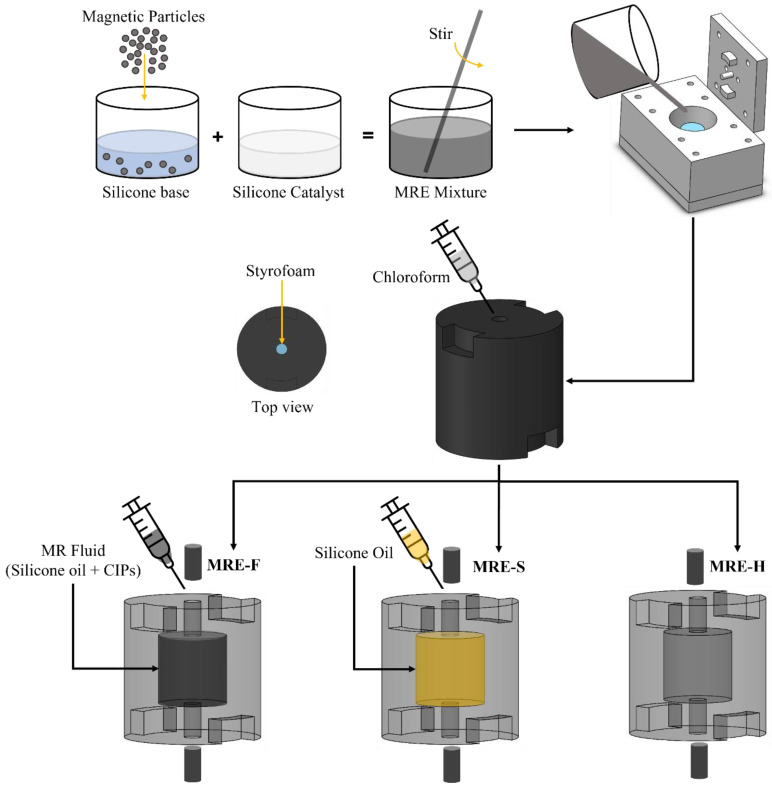
Schematic of the fabrication of the hybrid MRE materials.

**Figure 5 materials-15-03238-f005:**
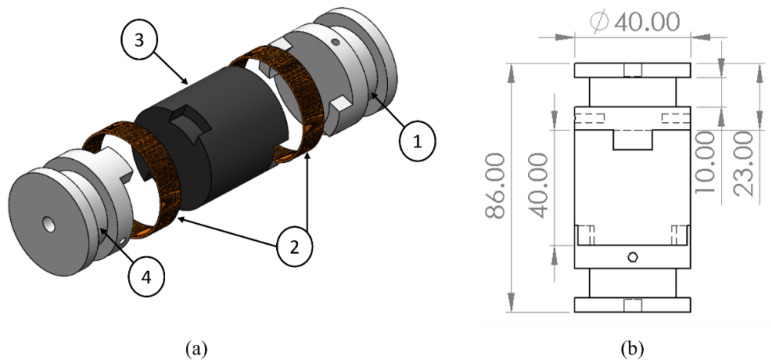
(**a**) Exploded view: 1—upper coupling hub; 2—coils; 3—MRE-layer; 4—lower coupling hub; (**b**) drawing view of MRE-based coupling (unit: in mm).

**Figure 6 materials-15-03238-f006:**
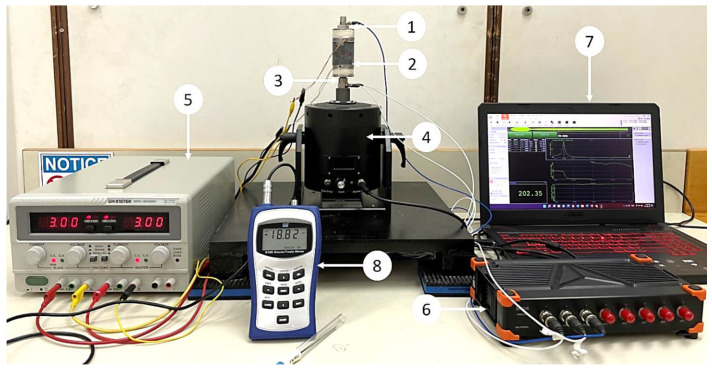
Experimental setup: (1) accelerometer, (2) hybrid MRE-based coupling, (3) impedance sensor, (4) vibration shaker, (5) regulated power supply, (6) data acquisition system, (7) computer with signal analyzing software, and (8) tesla meter.

**Figure 7 materials-15-03238-f007:**
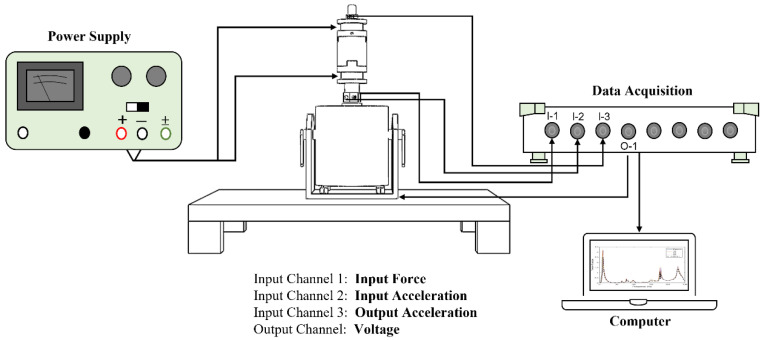
Schematic representation of the experimental setup.

**Figure 8 materials-15-03238-f008:**
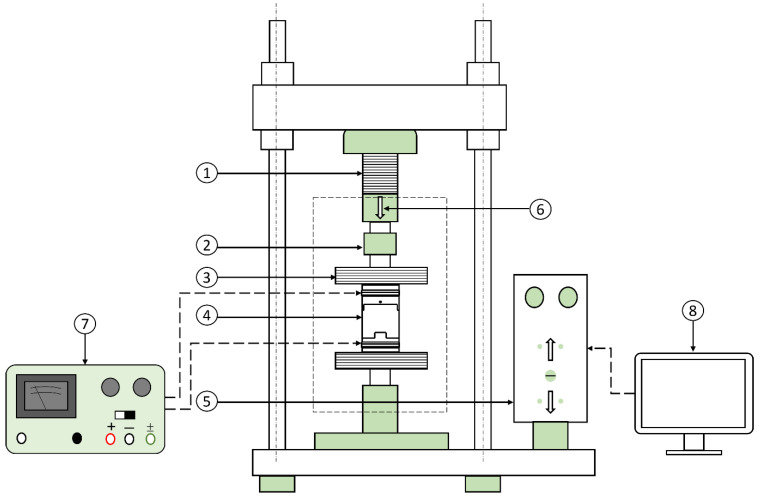
Schematic diagram of the compression testing machine: (1) moving stroke, (2) load cell, (3) wooden compression plates, (4) hybrid MRE-coupling, (5) position control unit, (6) compression direction, (7) power supply, and (8) computer.

**Figure 9 materials-15-03238-f009:**
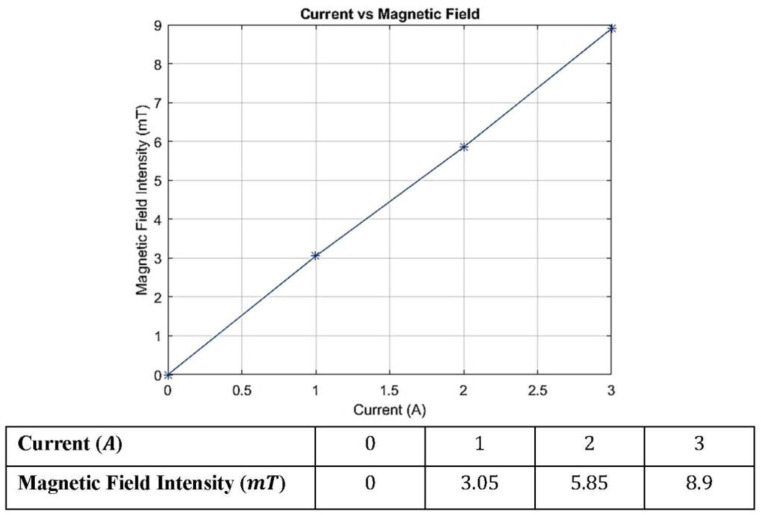
The magnetic field generated within the electromagnetic coils against the supplied current.

**Figure 10 materials-15-03238-f010:**
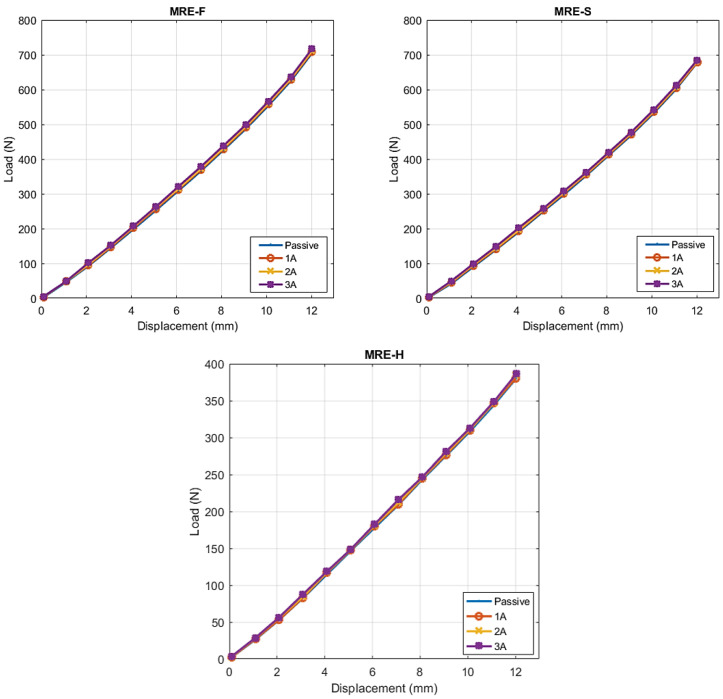
Load-displacement curves for MRE-F, MRE-S, and MRE-H.

**Figure 11 materials-15-03238-f011:**
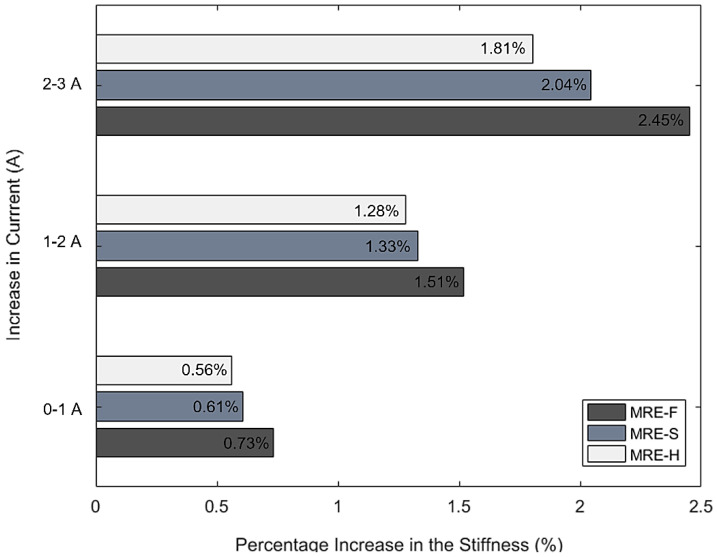
Percentage increase in stiffness for MRE-F, MRE-S, and MRE-H.

**Figure 13 materials-15-03238-f013:**
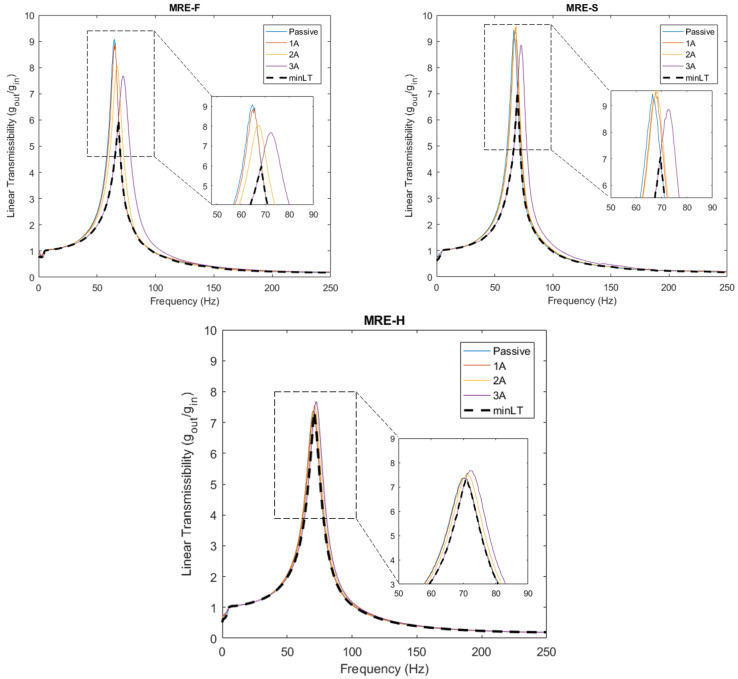
The linear transmissibility factor for MRE-F, MRE-S, and MRE-H.

**Figure 14 materials-15-03238-f014:**
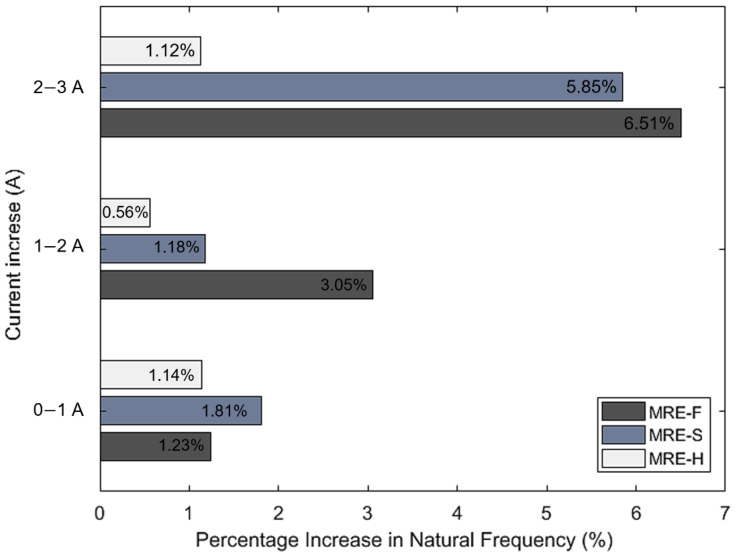
Percentage increase in natural frequency for MRE-F, MRE-S, and MRE-H.

**Figure 15 materials-15-03238-f015:**
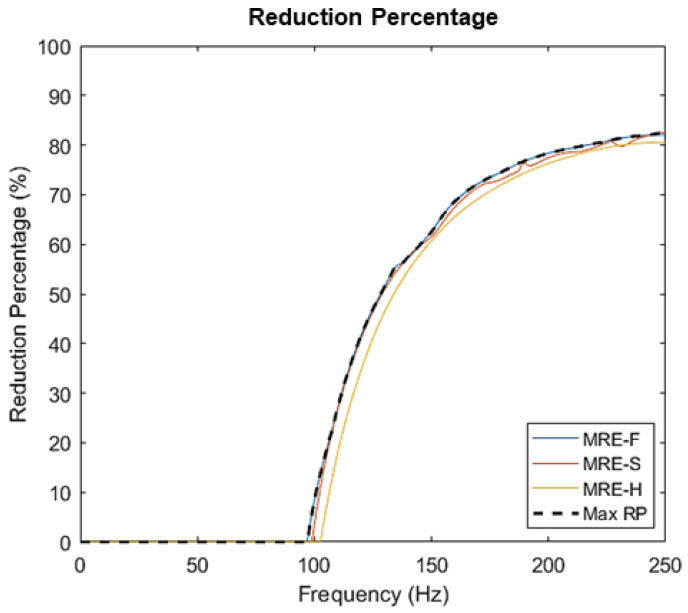
The reduction percentage curve for MRE-F, MRE-S, and MRE-H.

**Figure 16 materials-15-03238-f016:**
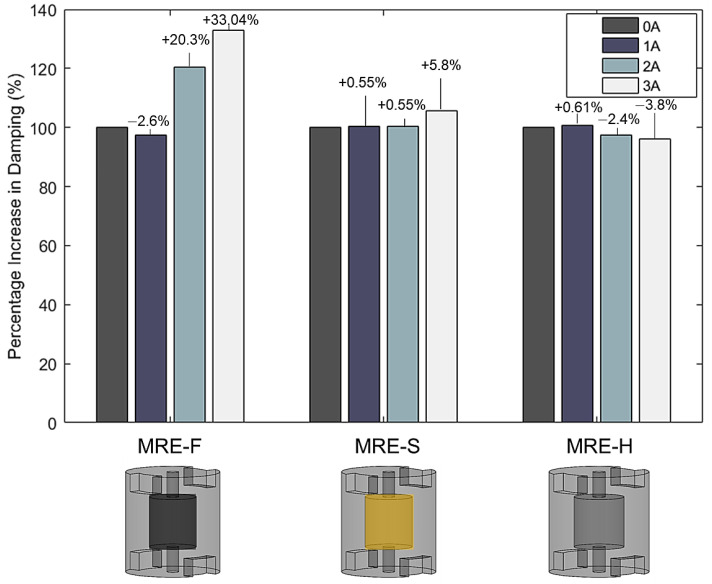
The percentage increase in the damping coefficient for MRE-F, MRE-S, and MRE-H.

**Table 1 materials-15-03238-t001:** Material properties of the silicone elastomer and CIPs.

Materials	Properties	
Silicone elastomer	$\mathrm{Density}\;(\mathrm{kg}/m^{3})$	1.06
	Hardness	22
	$\mathrm{Tear}\;\mathrm{resistance}\;(N/{\mathrm{mm}}^{2})$	5
	Elastic recovery (%)	99.95%
CIPs	Type	Carbonyl Iron—SQ-I
	$\mathrm{Density}\;(\mathrm{kg}/m^{3})$	7.89
	Particle size (μm)	4.5
	Coating	${\mathrm{SiO}}_{2}$
	$\mathrm{Permeability}\;(N/{\mathrm{mm}}^{2})$	10

**Table 2 materials-15-03238-t002:** The naming of the fabricated hybrid MRE samples.

SN.	Name	Elastomer	Carrier Fluid
1	MRE-F	MRE	Silicone oil + CIPs
2	MRE-S	MRE	Silicone oil
3	MRE-H	MRE	Hollow

**Table 3 materials-15-03238-t003:** Measured stiffness of MRE-F, MRE-S, and MRE-H at different current values.

Applied Current (A)	Stiffness (N/mm)		
	MRE-F	MRE-S	MRE-H
0	54.573	52.519	30.399
1	54.973	52.837	30.569
2	55.401	53.218	30.788
3	55.911	53.592	30.948

**Table 4 materials-15-03238-t004:** The zero-field modulus, magneto-induced modulus, and relative MR effect of MRE-F, MRE-S, and MRE-H.

Sample	Relative MR Effect			
	$E_{0}\;\left(MPa\right)$	$E_{max}\;\left(MPa\right)$	$\Delta E=E_{max}-E_{0}$	MR Effect (%)
MRE-F	1.866	1.898	0.032	1.72%
MRE-S	1.793	1.813	0.02	1.10%
MRE-H	1.006	1.017	0.011	1.095%

**Table 5 materials-15-03238-t005:** The damping coefficients of MRE-F, MRE-S, and MRE-H at different current values.

Applied Current (A)	$f_{1}$ (Hz)	$f_{2}$ (Hz)	$f_{n}$ (Hz)	Damping Ratio
**MRE-F**				
0	61.09	67.93	64.8	0.052833
1	61.46	68.19	65.6	0.051273
2	62.64	71.17	67.6	0.063098
3	67.20	77.19	72	0.069442
**MRE-S**				
0	63.09	69.88	69.4	0.051098
1	64.04	70.96	67.6	0.051226
2	64.29	71.26	68.4	0.050961
3	68.42	76.17	72.4	0.053521
**MRE-H**				
0	64.96	75.15	70	0.072807
1	65.06	75.40	70.8	0.073050
2	65.98	76.04	71.2	0.070604
3	67.20	77.19	72	0.069442

## Data Availability

The data presented in this study are available on reasonable request from the corresponding author.

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
