# Peer review of "Development of a Performance-Enhanced Hybrid Magnetorheological Elastomer-Fluid for Semi-Active Vibration Isolation: Static and Dynamic Experimental Characterization"

_materials, 2022, doi:10.3390/ma15093238_

Round 1

Reviewer 1 Report

The novelty of this work is in new kind of the magnetorheological elastomers, consisting of the host polymer with holes, either filled by magnetic fluids, either hollow.

The presented results show that the composites with the fluids are more rigid and demonstrate better damping characteristics than those with empty holes (as it should be, because of the fluid is viscous). 

Unfortunately, authors have not demonstrated the MR effects in these materials. Indeed, I have not found any dependence of the composite characteristics on magnetic field. Honestly speaking, this looks strange for the work, presenting new MR materials. No comparison of the mechanical properties of these composites with the properties of traditional MRE is given. Without that it is impossible to understand, what and were is the advantage of these composites over the traditional MRE with embedded magnetic particles. 

To my mind, the paper must be revised to demonstrate the MR effect in these systems and their potential advantage over the traditional MR composites.  

Minor remark. Above Fig.14 it should be "Reduction", not "Redcution".

Reviewer 2 Report

The paper deals with the design fabrication and characterization

of a magnetorheological elastomer enhanced with 

carbonil iron particles emersed in silicone oil.

The topic is of great interest to the audience,

it fits into the scope of the journal and it is

written in good manner.

I suggest publishing after some minor revisions

that would improve the scientific soundness:

Equations 3 and 4

TF Definition, X and Y should swap their places so that 

Y is up, in the nominator

X is down, in the denominator.

With many kind regards,

Reviewer 3 Report

On page 6 you mention: "The CIPs are added with a 10% volume fraction. This specific ratio is an intermediate between sufficient magnetization and passive stiffness". How you determined this and why 10% is an intermediate? Do you have some additional explanations for this?

You also mention that the size of CIP is 4.5 μm. Are CIPs monodisperse particles or this value is an average? How the size of CIPs and their size distribution could affect the performance of MRE-F?

A more detailed description of how the stiffness was measured from the slope of the load-displacement curves is needed (i.e. on which ranges and what are the errors in fitting parameters), since the figures show that  the curves are non-linear.

Does the stiffness changes with time? If yes, how?

English language shall be corrected, for example at page 1 you mention "Magnetorheological (MR) materials are smart composites due to their huge capabilities in many industrial applications". Obviously MR are not smart materials due to their capabilities in many industrial applications, but for other reasons.

Consistency within the text shall be checked once more. For example, what means gout and gin in the labels of Fig. 12?

The literature cited is not balanced. There are several well-known research groups with many papers on this topic which were not mentioned in the Introduction section. See for example the group of I. Bica et al. with papers describing the preparation of various hybrid MRE, their performance, and which also use mathematical modeling. 

Reviewer 4 Report

Dear Authors:

This work investigates the use of hybrid magnetorheological elastomer-fluid  for longitudinal vibration isolation. The hybrid MRE was fabricated by encapsulating MR fluid inside the elastomer matrix. Thus, this approach was shown to be effective in enhancing the MR effect of the elastomer by
providing a better response to the magnetic field, and  improving the vibration isolation properties. Based on results, authors showed that  hybrid MRE exhibited better stiffness change with current increase and recorded the highest value of 55.911 N/mm with better damping properties.

I think that the work is important and novel and a good contribution to literature. I request the acceptance of the manuscript with minor revison based on the following comments:

  1. The grammar of the manuscript, especially the abstract and the main text of the paper cotain errors. For this reason, authors should correct the grammar, and rewrite the main text of paper which contains the grammatical errors. For this reason, authors should check the grammar and the English of the paper with a Professional English language Institution .
  2. In the discussion part, authors discussed the general principles of what’s going on in the system in a very general sense. However, the discussion of these parts should be focused in more detail by giving citations from major works in this field.
  3. The introduction part should be written again by giving info from major highly cited works in the literature.

Round 2

Reviewer 3 Report

The Authors took into account my remarks.